# Vulnerable Forests of the Pink Sea Fan *Eunicella verrucosa* in the Mediterranean Sea

**Giovanni Chimienti** [1,2] 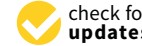

1   Dipartimento di Biologia, Università degli Studi di Bari, Via Orabona 4, 70125 Bari, Italy; giovanni.chimienti@uniba.it; Tel.: +39-080-544-3344
2   CoNISMa, Piazzale Flaminio 9, 00197 Roma, Italy

**Abstract:** The pink sea fan *Eunicella verrucosa* (Cnidaria, Anthozoa, Alcyonacea) can form coral forests at mesophotic depths in the Mediterranean Sea. Despite the recognized importance of these habitats, they have been scantly studied and their distribution is mostly unknown. This study reports the new finding of *E. verrucosa* forests in the Mediterranean Sea, and the updated distribution of this species that has been considered rare in the basin. In particular, one site off Sanremo (Ligurian Sea) was characterized by a monospecific population of *E. verrucosa* with $2.3 \pm 0.2$ colonies m$^{-2}$. By combining new records, literature, and citizen science data, the species is believed to be widespread in the basin with few or isolated colonies, and 19 *E. verrucosa* forests were identified. The overall associated community showed how these coral forests are essential for species of conservation interest, as well as for species of high commercial value. For this reason, proper protection and management strategies are necessary.

**Keywords:** Anthozoa; Alcyonacea; gorgonian; coral habitat; coral forest; VME; biodiversity; mesophotic; citizen science; distribution

## 1. Introduction

Arborescent corals such as antipatharians and alcyonaceans can form mono- or multispecific animal forests that represent vulnerable marine ecosystems of great ecological importance [1–4]. The order Alcyonacea includes the so-called gorgonians, i.e., octocorals characterized by a tough but flexible axis composed of the scleroprotein gorgonin [5]. Varying amounts of calcareous material is included in the axis as well as in the living tissue, where the calcareous structures take the form of sclerites [5]. The Mediterranean Sea hosts 24 recognized species of gorgonians, sometimes known as "sea fan" or "sea whip" depending on the shape of the colony. Gorgonian forests contribute greatly to the Mediterranean seascape, providing biomass, structural complexity, and aesthetic value to sublittoral communities, as well as sustaining a rich, associated biodiversity [6–14].

The pink sea fan *Eunicella verrucosa* (Pallas, 1766) is an Atlanto-Mediterranean species present in the East Atlantic, from Scotland to Angola, and in the Mediterranean Sea [5,15–17]. It is a temperate- to cold-water species that can be found from 2 to 60 m depth in the Atlantic Ocean [5,17], where it represents a major heritage species [18–20]. In the Mediterranean Sea, *E. verrucosa* has been listed as "near threatened" on the Red List by the International Union for Conservation of Nature (IUCN) [21], and has been reported from 20 to 200 m depth [5,22,23]. This species lives in the mesophotic zone, such as that portion of the seabed that goes from the limit of presence of seagrass to the limit of presence of algae (loss of net productivity at level of irradiance <1%), indicatively between 50 and 200 m depth [2,4]. This bathymetric range can host rocky pinnacles commonly included among the so-called *roche du large* [24], literally 'offshore rocks', representing the ideal substrate for *E. verrucosa*. Occasionally, it can also form *facies* (*sensu* [24]) on coralligenous formations, such as typical Mediterranean bioconstructions

built up by a suite of calcifying organisms (e.g., calcareous red algae, corals, serpulids, bryozoans, molluscs) that grow one on the other, generation after generation, building a secondary hard substratum in dim light conditions [13,25,26]. Although knowledge about the Mediterranean mesophotic zone is scarce and the distribution of *E. verrucosa* remains still unclear, recent records showed that the species is mostly present below 35 m depth [22,23]. Shallower occurrences are uncommon and most likely due to low-light conditions (e.g., water turbidity, presence of rocks and overhangs) [27] in presence of cold-enough waters. In particular, *E. verrucosa* has been found in the Alboran Sea, in the Balearic Sea, and in the Ligurian Sea, with few more records along the coasts of Tunisia, Italy, Croatia, and Greece [17,22,23,27–37]. All these records refer to the occurrence of one or few isolated colonies, sometimes as part of a mixed coral forest, while no data about Mediterranean forests of *E. verrucosa* are present in literature. The only population investigated in this basin is located off Marseille, in the Gulf of Lion (Balearic Sea), where 73 colonies distributed within nine sites were monitored [22]. However, it is not clear if the area is characterized by a dense coral assemblage or by several isolated colonies over a large area, since colonies' density is not reported.

This study reports: (1) The remarkable new finding of *E. verrucosa* forests in the Mediterranean Sea, (2) an overview about the currently known distribution of the species in the basin, and (3) a first insight about the biodiversity associated with these coral forests.

## 2. Materials and Methods

Technical scuba dives were carried out in two sites located off the cities of Bordighera and Sanremo (Ligurian Sea, Italy) in order to verify the presence of *E. verrucosa* according to informal reports from local divers. Taxonomic identification of *E. verrucosa* was based on macroscopic morphological characters, such as the pink or white color, the fan shape with many branches on one plant, the thin terminal branches, and the high calyces arranged biserially [5]. These characters allowed distinguishing the species from the two congeneric fan-shaped ones: *Eunicella cavolini* (yellow color, terminal branches cylindrically, and calyces relatively low and disposed all around the branches) and *Eunicella singularis* (white, greyish, or greenish color, long terminal branches directing upright, and calyces not projecting) [5].

A shoal off Sanremo, called "Scoglio dell'Astice", was selected as study site. Underwater videos were carried out using a 4K camera with a maximum scene illuminator of 10,000 lumen and a size reference to calculate the area investigated. Video analysis was performed using Adobe Premiere Pro software, and sampling units of $2.5 \pm 0.2$ m$^2$ were defined along each transect, according to the minimal area used for visual surveys on octocoral habitats [38–40]. Sequences with bad visibility and/or recorded outside the coral forest (e.g., soft bottoms, hard-bottom areas not colonized by *E. verrucosa*) were discarded. The forest of *E. verrucosa* was quantified by abundance (number of colonies per sampling unit), then the density (colonies m$^{-2}$) was calculated for each sampling unit and expressed as mean ± standard error. Anthropic impacts observed within the coral forest were also assessed.

High-resolution photos of *E. verrucosa* colonies with size reference were collected in the same coral forest for nondestructive morphometric analysis. Twelve size classes of 5 cm each were identified, from 1–5 to 56–60 cm. Size structure was analyzed in terms of size-frequency and distribution parameters, such as skewness and kurtosis, calculated using the R software platform functions *agostino.test* [41] and *anscombe.test* [42]. Then, the age of the colonies was inferred, based on the height (H), according to the function proposed by [22], as follow:

$$\text{Age} = \text{e}^{\frac{\text{H}+18.39}{17.94}} \qquad (1)$$

Age structure of the population was analyzed in the same way as the size structure, based on 15 age classes of 5 years each, from 1–5 to 71–75 years.

The general distribution of *E. verrucosa* in the Mediterranean Sea was assessed through a multisource approach based on new records, literature, and citizen science data. In particular,

additional records were obtained from grab and rock-dredge samples collected along the coasts of Montenegro (Adriatic Sea, cruise CROMA—*CROatian and Montenegrin mArine ecosystems*) and Greece (Ionian Sea; cruise COCOMAP14—*COast to COast habitat MAPping 2014*), aboard Research Vessel *Urania*. Data present in scientific literature were also considered [22,23,27,29–37,43,44], and only a few uncertain records were discarded (e.g., [28,45]). Citizen science data available online were used only in the presence of good-quality photos to confirm the specific identification or if they were previously validated by an expert. In particular, data were extrapolated from three main platforms: iNaturalist (www.inaturalist.org, a social network of nature enthusiasts that share and cross-validate photographic observations), the Global Biodiversity Information Facility (www.gbif.org, an international network to share open access data about life), and the open access database of the Reef Check Mediterranean Underwater Coastal Environment Monitoring protocol (www.reefcheckmed.org, that groups' underwater observations were carried out by trained volunteers based on a standardized protocol [46]). This latter platform includes a semi-quantitative estimation of abundances. Therefore, the observations of more than 50 colonies of *E. verrucosa* during a single dive were considered as coral forests.

The macro- and megafauna associated with *E. verrucosa* forests was identified at the lowest possible taxonomic level based on images and samples collected in this study. Observations from literature were also considered, particularly for epibionts [5,22,31,47]. Epibiont species, as well as species of conservation importance and of commercial value, were highlighted.

## 3. Results

### 3.1. The Forest of Eunicella verrucosa off Sanremo

The shoal explored off Sanremo (Scoglio dell'Astice) was composed by two main rocky areas, distant 50 m one from the other and surrounded by muddy sediments. The first area was characterized by an elevation of up to 4 m and steep walls, over an area of approximatively 200 m$^2$ entirely colonized by a monospecific forest of *Paramuricea clavata*, with numerous and large colonies mostly settled on the top and on the edge of the small shoal, at 65–68 m depth. Numerous massive sponges were also observed (Figure 1a,b). The second area was represented by scattered large rocks and small rocky platforms of up to 1.5 m in height over an area of about 120 m$^2$, inhabited by a monospecific forest of *E. verrucosa* at 65–70 m depth (Figure 1c–e). A total of 158 colonies of *E. verrucosa* were counted over an area of 80 m$^2$, with a mean density of 2.3 ± 0.2 colonies m$^{-2}$ (28 sampling units, 1 to 13 colonies per sampling unit) and a maximum of 5.2 colonies m$^{-2}$. The understory of the coral forest was dominated by sponges of the genus *Axinella*, often colonized by the zoanthid *Parazoanthus axinellae* (Figure 1d). Two colonies of *P. clavata* and two of *E. cavolini* were also present in the highest portion of the rocky shoal. The nearby area was characterized by a muddy seabed with numerous holothurians, as well as small scattered rocky substrates and coralligenous, patchy bioconstructions dominated by sponges and ascidians. Part of the nearby hard bottom was also covered by countless specimens of the polychaete *Sabella pavonina* (Figure 1f,g).

All the colonies of *E. verrucosa* belonged to the white morphotype *sensu* [5]. Colonies' sizes ranged from 6 to 58 cm in height, with the dominant presence of the size classes from 21 to 35 cm (82 colonies measured). Size-frequency distribution resulted symmetrically and mesokurtically, with the mean, median, and modal values overlapping (Figure 2).

The inferred age of the colonies was from 4 to 71 years, showing a highly skewed and leptokurtic distribution (Figure 3), with a long right tail that represented colonies older than 20 years and generally higher than 35 cm.

Anthropic impacts consisted of few lost fishing gears, such as two bottom longlines and one trammel net, showing that this area has been occasionally targeted by small artisanal fishery.

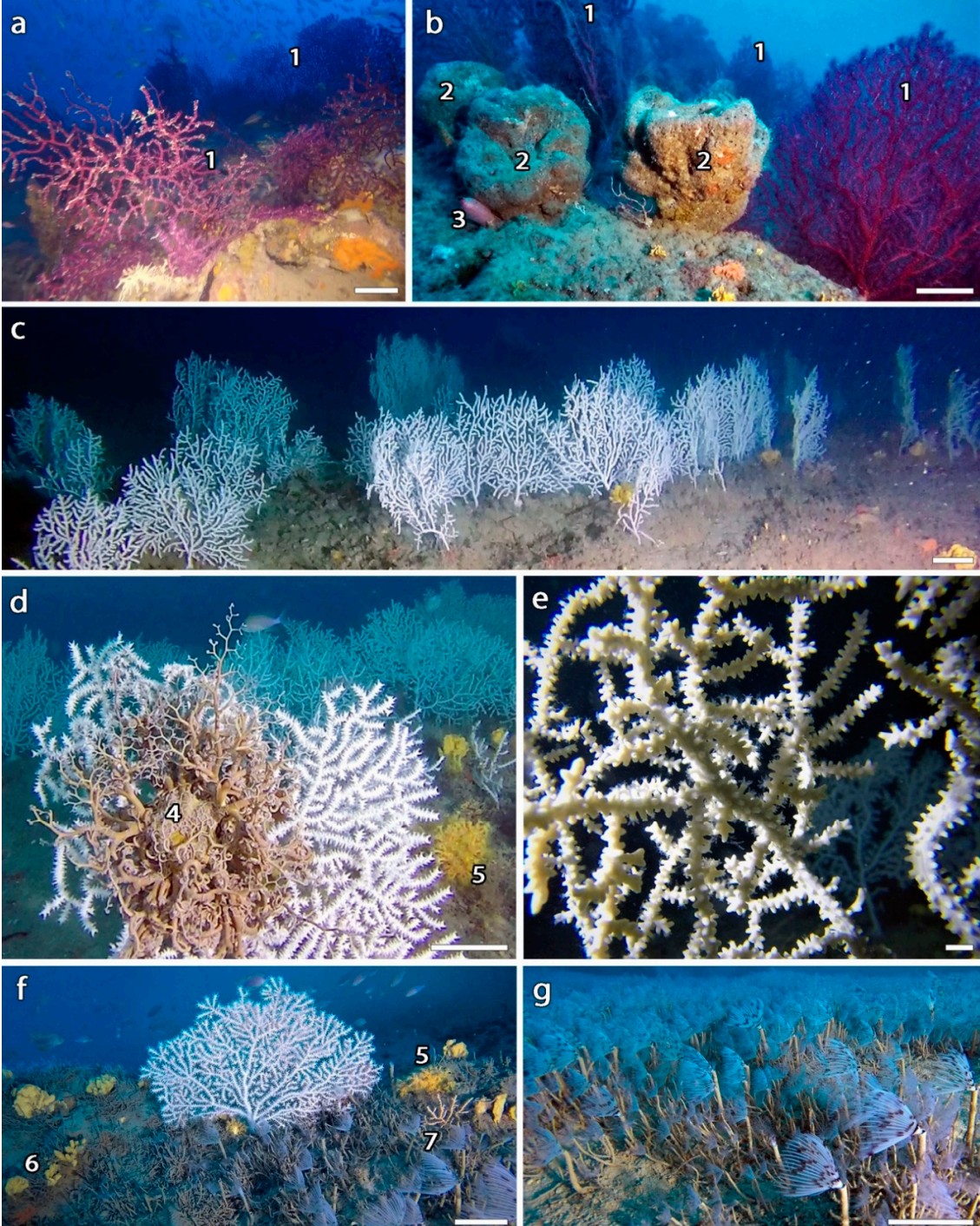

**Figure 1.** Benthic communities off Sanremo. (**a**) Forest of *Paramuricea clavata* (1), with (**b**) massive sponges (2) and the ascidian *Halocynthia papillosa* (3); (**c**) forest of *Eunicella verrucosa*, with (**d**) the epibiont ophiuroid *Astrospartus mediterraneus* (4), and the sponges *Axinella* sp. covered by *Parazoanthus axinellae* (5); (**e**) living polyps of *E. verrucosa*; (**f**) isolated colony of *E. verrucosa* on the rocky bottom nearby the coral forest, whose community is locally dominated by *Axinella* spp. with (5) and without (6) *P. axinellae*, as well as the polychaete *Sabella pavonina* (7); (**g**) detail of *S. pavonina* population. Scale bars: a–d,f,g, 10 cm; e, 1 cm.

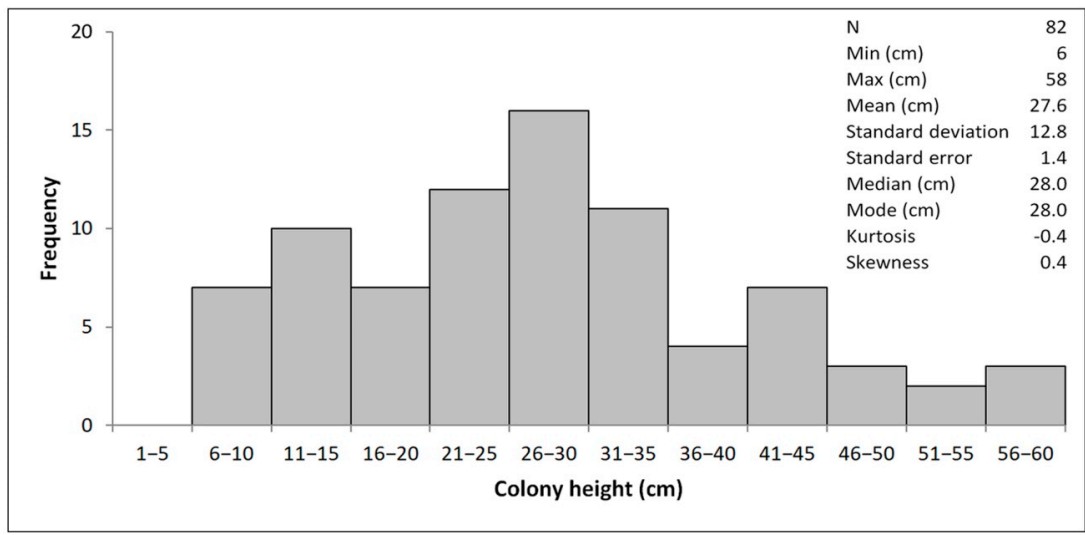

**Figure 2.** Size-frequency distribution of *Eunicella verrucosa* off Sanremo.

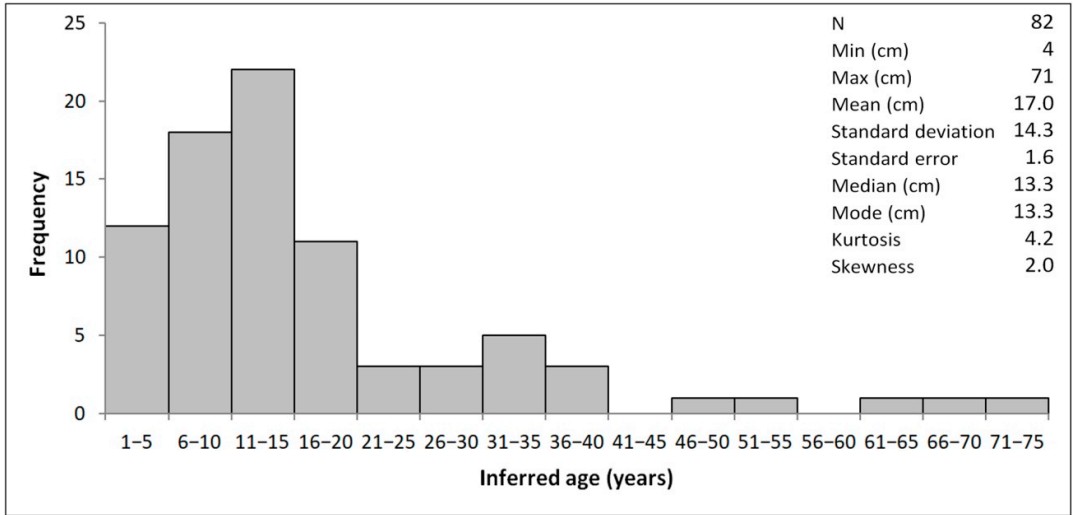

**Figure 3.** Age-frequency distribution of *Eunicella verrucosa* off Sanremo.

### 3.2. The Coral Forest off Bordighera

A mixed coral forest, dominated by numerous, large colonies of *E. verrucosa* and *P. clavata*, with the occasional occurrence of *E. cavolinii* and *Leptogorgia sarmentosa*, was found on a rocky shoal off Bordighera, at 47–50 m depth (Figure 4a). Most of the *E. verrucosa* colonies showed the white morphotype, although some pink colonies were also present (Figure 4b). A second rocky shoal was detected in the area, a few hundred meters from the first one, with a mixed coral assemblage dominated by *P. clavata* and the black coral *Antipathella subpinnata* at 68–70 m depth (Figure 4c).

### 3.3. Additional Records of Eunicella verrucosa in the Adriatic and Ionian Seas

Several *E. verrucosa* colonies and branches were collected off and southeast of the Kotor Bay through grab and dredge sampling, respectively (Figure 4d,e). Even if it was not possible to quantify the abundance, the presence of two forests in the two sampling sites was assumed based on the large amount of sampled material. All the colonies in this area were pink in color. Two additional pink colonies of *E. verrucosa* were collected by rocky dredge off Merlera Island (Greece, Ionian Sea), at 90 m depth.

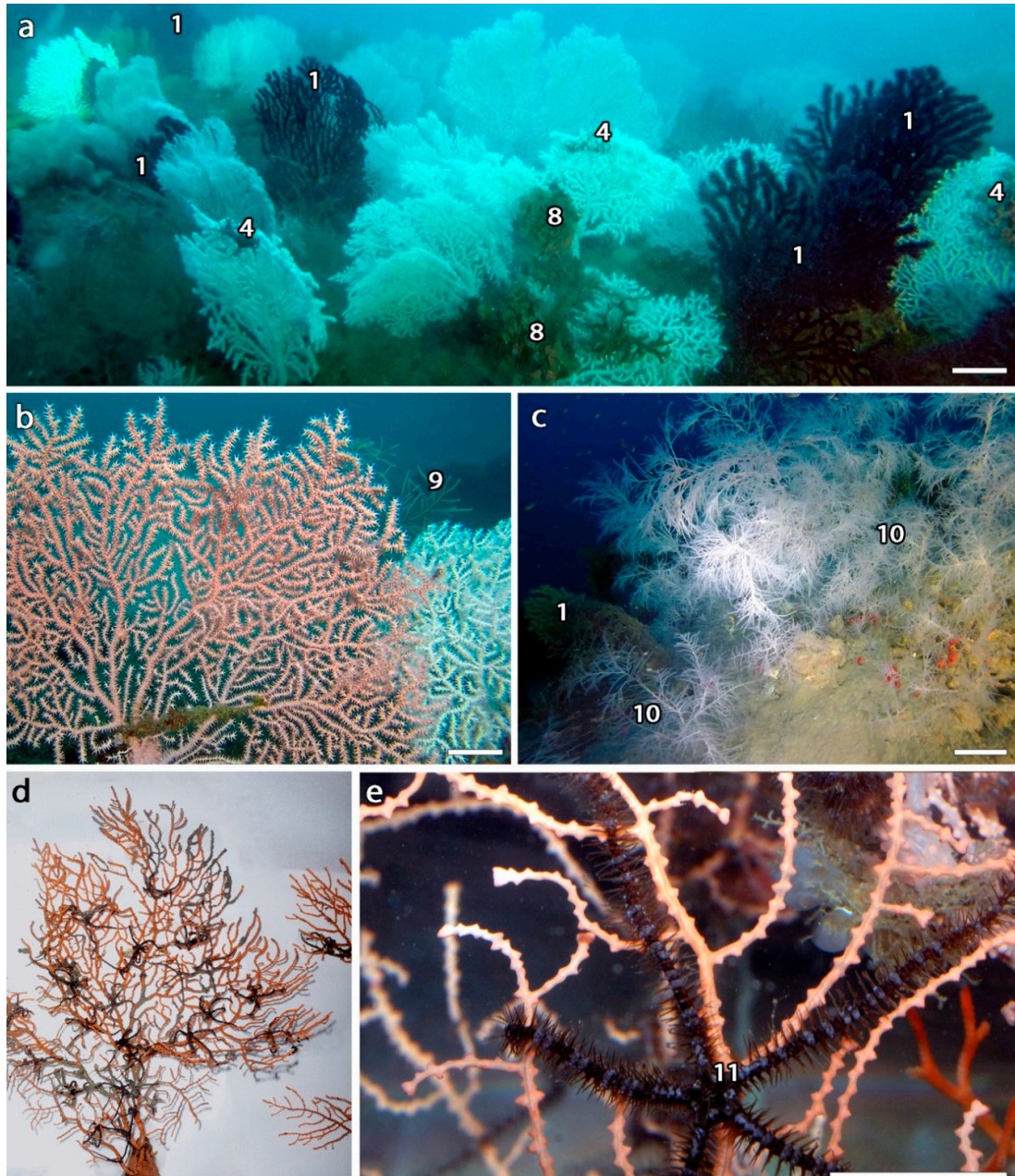

**Figure 4.** Additional records of *Eunicella verrucosa*. (**a**) Mixed forest of *E. verrucosa* and *Paramuricea clavata* (1) off Bordighera, with colonies epibionted by the ophiuroid *Astrospartus mediterraneus* (4) and by bryozoans (8); (**b**) detail of both pink and white morphotypes of *E. verrucosa*; in the background, a colony of *Leptogorgia sarmentosa* (9); (**c**) mixed forest of *P. clavata* (1) and *Antipathella subpinnata* (10) off Bordighera; (**d**) colonies of *E. verrucosa* sampled southeast of Kotor, with many epibiont ophiuroids *Ophiacantha setosa*; (**e**) detail of the colony with *O. setosa* (11). Scale bars: a,c, 10 cm; b,d–e, 5 cm.

## 3.4. Distribution of Eunicella verrucosa in the Mediterranean Sea

The combination of new data from visual surveys, samples, scientific literature, and citizen science allowed obtaining an updated overview about the distribution of *E. verrucosa* in the Mediterranean Sea (Figure 5a). The species was particularly widespread in the northwestern portion of the basin, along the coasts of Spain, France, and Italy. The majority of the records concerned the occurrence of isolated or few colonies, identified in almost 162 sites (1 in this study, 22 from literature, and 139 from citizen science), while forests of *E. verrucosa* have been observed in 19 sites thus far (4 in this study, 1 from

literature, and 14 from citizen science), most of which were in the Ligurian Sea and Sicily Channel (Figure 5b, Table 1). Globally, *E. verrucosa* was mostly present below 30 m depth, with some records at shallower depths reported in the Sicily Channel (the shallowest at 16 m depth).

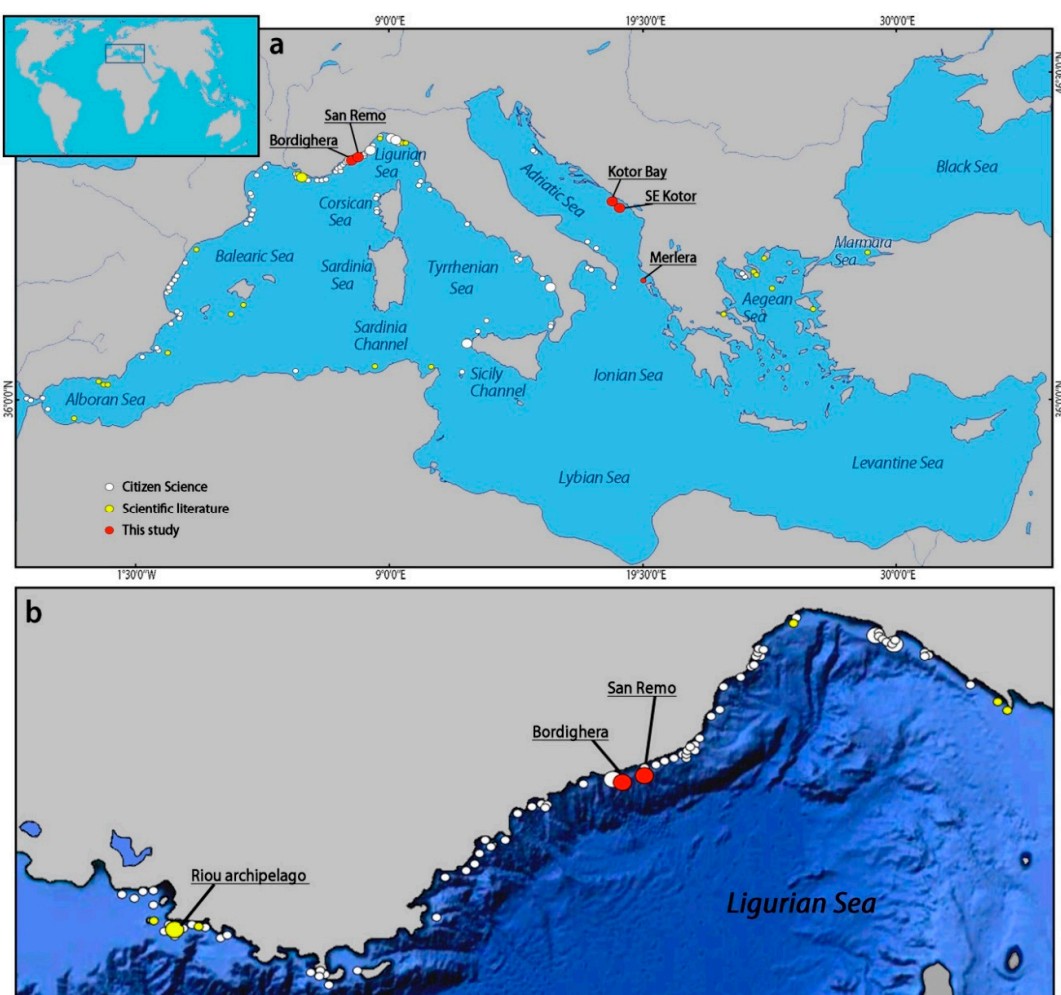

**Figure 5.** Distribution of *Eunicella verrucosa* in the Mediterranean Sea. (**a**) Global view, (**b**) detail of the Liguro–Provencal coast. Small dots indicate few/isolated colonies, while large dots represent forests of *E. verrucosa*.

**Table 1.** Localization, depth (m), and abundance (N: number of colonies) of the 19 *Eunicella verrucosa* forests identified in the Mediterranean Sea, with indication of the latest update and the source of the data. Fr: France; It: Italy; Mo: Montenegro; RCMed: Reef Check Mediterranean.

| Basin | Country | Locality | Lat N | Lon E | N | Depth | Year | Source |
|---|---|---|---|---|---|---|---|---|
| Balearic Sea | Fr | Maire Island, Riou Archipelago | 43.2076 | 5.3402 | - | 20–40 | 2007 | [22] |
| Ligurian Sea | It | I tuvi Bordighera | 43.7691 | 7.6878 | >50 | 30–33 | 2012 | RCMed |
| Ligurian Sea | It | Bordighera | 43.768 | 7.694 | >50 | 47–50 | 2020 | This study |
| Ligurian Sea | It | Sanremo, Scoglio dell'Astice | 43.800 | 7.838 | >150 | 65–70 | 2020 | This study |
| Ligurian Sea | It | Pilone | 43.9158 | 8.1404 | >50 | 33–35 | 2014 | RCMed |
| Ligurian Sea | It | Portofino, Punta Chiappa | 44.3230 | 9.1456 | >50 | 45–55 | 2006 | RCMed |
| Ligurian Sea | It | Portofino, Punta del Faro | 44.2976 | 9.2181 | >50 | 40–70 | 2015 | RCMed |

| Basin | Country | Locality | Lat N | Lon E | N | Depth | Year | Source |
|---|---|---|---|---|---|---|---|---|
| Ligurian Sea | It | Portofino, Punta del Faro | 44.2980 | 9.2183 | >50 | 42–45 | 2013 | RCMed |
| Ligurian Sea | It | Portofino, Punta del Faro | 44.2980 | 9.2196 | >50 | 45–50 | 2013 | RCMed |
| Ligurian Sea | It | Punta Manara | 44.2427 | 9.4027 | >50 | 40–45 | 2006 | RCMed |
| Tyrrhenian Sea | It | Scalea | 39.8177 | 15.7731 | >100 | 44–55 | 2010 | RCMed |
| Sicily Channel | It | Favignana | 37.9048 | 12.3059 | >50 | 25–30 | 2010 | RCMed |
| Sicily Channel | It | Favignana, Trigone | 37.9068 | 12.3037 | >50 | 25–30 | 2010 | RCMed |
| Sicily Channel | It | Favignana, Galeotta | 37.9114 | 12.2999 | >50 | 18–30 | 2010 | RCMed |
| Sicily Channel | It | Favignana, Toro Canyon | 37.8771 | 12.3095 | >100 | 16–32 | 2010 | RCMed |
| Sicily Channel | It | Favignana, Secca del Toro | 37.8782 | 12.3097 | >100 | 20–30 | 2010 | RCMed |
| Sicily Channel | It | Favignana, Secca Fondale | 37.8744 | 12.3091 | >100 | 20–30 | 2010 | RCMed |
| Adriatic Sea | Mo | Kotor Bay | 42.3015 | 18.5127 | - | 108 | 2014 | This study |
| Adriatic Sea | Mo | SE Kotor | 42.2312 | 18.5919 | - | 110 | 2014 | This study |

*3.5. Biodiversity Associated with Eunicella verrucosa Forests*

The overall macro- and megafauna observed in association with Mediterranean *E. verrucosa* forests included 80 taxa, i.e., 1 Foraminifera, 20 Porifera, 9 Cnidaria, 4 Mollusca, 5 Annelida, 4 Arthropoda, 7 Bryozoa, 12 Echinodermata, and 18 Chordata (Table 2), most of which were observed within the study site off Sanremo.

The sessile fauna was mainly composed by hard-bottom filter-feeders, such as sponges and ascidians, as well as suspension-feeders such as corals, polychaetes and bryozoans. Sponges were dominant in terms of diversity and represented a common component of the forests' understory, although many taxa were not identifiable based on images. The co-occurrence of *E. verrucosa* with other gorgonians, such as *E. cavolini*, *P. clavata,* and *L. sarmentosa,* was quite common, at least in the forests of the Liguro-Provencal coast (Figure 6a, Table 2).

Vagile and sedentary fauna included several echinoderms, molluscs, crustaceans, and fish. Echinoderms had representatives from all the five classes (Crinoidea, Holothuroidea, Asteroidea, Ophiuroidea, and Echinoidea), with holothurians commonly present in forests settled on short and scantly structured rocky outcrops, partially covered by sediments, that were particularly accessible to these soft-bottom deposit-feeders (Figure 6a,g). Mollusks included corallivores that feed on *E. verrucosa*, such as the Tritoniidae *Tritonia nilsodhneri* and the Ovulidae *Simnia spelta* (Figure 6c–d), as well as the spongivore *Felimare picta* (Figure 6e) that feeds on the numerous sponges, particularly those belonging to the genera *Ircinia* and *Dysidea*. Most of the fish fauna observed was represented by demersal species, such as the small red scorpionfish *Scorpaena notata*, the comber *Serranus cabrilla*, the rainbow wrasse *Coris julis,* and the cuckoo wrasse *Labrus mixtus*, as well as large schools of the red mullet *Mullus barbatus barbatus,* the swallowtail seaperch *Anthias anthias,* and the parrot seaperch *Callanthias ruber*. Large predators, such as the red scorpionfish *Scorpaena scrofa* (Figure 6f), the conger eel *Conger conger*, the Mediterranean moray eel *Muraena helena* (Figure 6g), the forkbeard *Phycis phycis*, the dusky grouper *Epinephelus marginatus,* and the Atlantic wreckfish *Polyprion americanus,* were also spotted off Sanremo. Schools of pelagic species, such as picarels *Spicara* sp. and horse mackerels *Trachurus* sp., were also observed swimming close to the coral forest.

Epibionts included benthic species such as the colonial polychaetes belonging to *Filograna–Salmacina* complex, the bivalve *Pteria hirundo*, and the bryozoan *Reteporella* cf. *grimaldii* (Figure 6b), whose settlement was probably enhanced by the grazing activities of the above-mentioned corallivores. These latter could be considered vagile epibionts, together with the crustacean *Balssia*

*gasti*, the crinoid *Antedon mediterranea* and the ophiuroids *Astrospartus mediterraneus* and *Ophiacantha setosa* (Figure 1d, Figure 4a, Figure 6a). Crinoids and ophiuroids have been observed using the colonies of *E. verrucosa* to lift from the seabed and enhance their chances to catch the suspended food particles. In particular, *O. setosa* was found with a large number of specimens on the colonies collected off and southeast of the Bay of Kotor (Figure 4d,e).

**Table 2.** List of the animal taxa observed in association with *Eunicella verrucosa* and its forests, with indication of the location and the protection framework. B: Bern Convention (Convention on the Conservation of European Wildlife and Natural Habitats: Appendix II and III), S: SPA/BD Protocol (Protocol for Specially Protected Areas and Biological Diversity in the Mediterranean, Barcelona Convention: Annex II and III), I: International Union for Conservation of Nature (IUCN), Red List (LC: least concern, NT: near threatened, VU: vulnerable).

| Phylum, Class | Taxon | Location | Protection | | |
|---|---|---|---|---|---|
| | | | B | S | I |
| **Foraminifera** | | | | | |
| Globothalamea | *Miniacina miniacea* (Pallas, 1766) | SE Kotor | | | |
| Porifera Demospongiae | *Agelas oroides* (Schmidt, 1864) | Sanremo | | | |
| | *Aplysina aerophoba* (Nardo, 1833) | Sanremo | | II | |
| | *Axinella brondstedi* Bergquist, 1970 | Bordighera, Sanremo | | | |
| | *Axinella damicornis* (Esper, 1794) | Sanremo | | | |
| | *Axinella polypoides* Schmidt, 1862 | Bordighera, Sanremo | II | II | |
| | *Chondrosia reniformis* Nardo, 1847 | Sanremo | | | |
| | *Cliona* sp. | Riou [22], Bordighera | | | |
| | *Dysidea avara* (Schmidt, 1862) | Bordighera | | | |
| | *Geodia* sp. | Sanremo | | | |
| | *Hemimycale columella* (Bowerbank, 1874) | Sanremo | | | |
| | *Ircinia* sp. | Sanremo | | | |
| | *Pachastrella monilifera* Schmidt, 1868 | Sanremo | | | |
| | *Petrosia* sp. | Bordighera, Sanremo | | | |
| | *Phorbas* sp. | Sanremo | | | |
| | *Pleraplysilla spinifera* (Schulze, 1879) | Sanremo | | | |
| | *Raspaciona aculeata* (Johnston, 1842) | Bordighera | | | |
| | *Sarcotragus* cf. *foetidus* Schmidt, 1862 | Sanremo | | II | |
| | *Spongia* sp. | Sanremo | | | |
| | Suberitidae | Sanremo | | | |
| Homoscleromorpha | *Oscarella lobularis* (Schmidt, 1862) | Sanremo | | | |
| Cnidaria Hydrozoa | *Aglaophenia* sp. | Sanremo | | | |
| | *Halecium* sp. | SE Kotor | | | |
| Anthozoa | *Alcyonium acaule* Marion, 1878 | Sanremo | | | LC |
| | *Caryophyllia* sp. | Kotor Bay, SE Kotor, Bordighera, Sanremo | | | |
| | *Eunicella cavolini* (Koch, 1887) | Riou [22], Bordighera, Sanremo | | | NT |
| | *Leptogorgia sarmentosa* (Esper, 1789) | Riou [22], Bordighera | | | LC |
| | *Paramuricea clavata* (Risso, 1826) | Riou [22], Bordighera, Sanremo | | | VU |
| | *Parazoanthus axinellae* (Schmidt, 1862) | Bordighera, Sanremo | | | LC |
| | *Savalia savaglia* (Bertoloni, 1819) | Sanremo | II | II | NT |
| Mollusca Gastropoda | *Felimare picta* (Philippi, 1836) | Sanremo | | | |
| | *Simnia spelta* (Linnaeus, 1758) [1] | Riou [22], SE Kotor | | | |
| | *Tritonia nilsodhneri* Marcus Ev., 1983 [1] | Grananda [47], Riou [22], Portofino [31] | | | |
| Bivalvia | *Pteria hirundo* (Linnaeus, 1758) [1] | SE Kotor | | | |

**Table 2.** *Cont*.

| Phylum, Class | Taxon | Location | Protection | | |
|---|---|---|---|---|---|
| | | | B | S | I |
| Annelida | | | | | |
| Polychaeta | *Bonellia viridis* Rolando, 1822 | Bordighera, Sanremo | | | |
| | *Filograna-Salmacina* complex [1] | Bordighera, Sanremo | | | |
| | *Sabella pavonina* Savigny, 1822 | Sanremo | | | |
| | *Serpula vermicularis* Linnaeus, 1767 | Sanremo | | | |
| | Serpulidae | Sanremo, SE Kotor | | | |
| Arthropoda | | | | | |
| Malacostraca | *Balssia gasti* (Balss, 1921) [1] | Riou [22] | | | |
| | *Homarus gammarus* (Linnaeus, 1758) | Sanremo | III | III | LC |
| | *Maja squinado* (Herbst, 1788) | Sanremo | III | III | |
| | *Palinurus elephas* (Fabricius, 1787) | Sanremo | III | III | VU |
| Bryozoa | | | | | |
| Gymnolaemata | *Adeonella calveti* Canu & Bassler, 1930 | Sanremo | | | |
| | *Myriapora truncata* (Pallas, 1766) | Sanremo | | | |
| | *Pentapora fascialis* (Pallas, 1766) | Sanremo | | | |
| | *Reteporella* cf. *grimaldii* (Jullien, 1903) [1] | Bordighera, Sanremo | | | |
| | *Schizomavella (Schizomavella) mamillata* (Hincks, 1880) | Sanremo, Kotor Bay | | | |
| | *Schizomavella* sp. | Sanremo | | | |
| | *Smittina cervicornis* (Pallas, 1766) | Sanremo | | | |
| Echinodermata | | | | | |
| Crinoidea | *Antedon mediterranea* (Lamarck, 1816) [1] | Kotor Bay, SE Kotor | | | |
| Holothuroidea | *Holothuria (Holothuria) tubulosa* Gmelin, 1791 | Bordighera, Sanremo | | | LC |
| | *Holothuria (Panningothuria) forskali* Delle Chiaje, 1823 | Bordighera, Sanremo | | | LC |
| | *Holothuria (Roweothuria) poli* Delle Chiaje, 1824 | Bordighera, Sanremo | | | LC |
| Asteroidea | *Anseropoda placenta* (Pennant, 1777) | SE Kotor | | | |
| | *Echinaster (Echinaster) sepositus* (Retzius, 1783) | Sanremo | | | |
| | *Hacelia attenuata* Gray, 1840 | Sanremo | | | |
| | *Marthasterias glacialis* (Linnaeus, 1758) | Sanremo | | | |
| | *Peltaster placenta* (Müller & Troschel, 1842) | Sanremo | | | |
| Ophiuroidea | *Astrospartus mediterraneus* (Risso, 1826) [1] | Bordighera, Sanremo | | | |
| | *Ophiacantha setosa* (Bruzelius, 1805) [1] | Corsica [5], Kotor Bay, SE Kotor | | | |
| Echinoidea | *Echinus melo* Lamarck, 1816 | Sanremo | | | |
| Chordata | | | | | |
| Ascidiacea | Didemnidae | Sanremo, Kotor Bay, SE Kotor | | | |
| | *Diplosoma spongiforme* (Giard, 1872) | Sanremo | | | |
| | *Halocynthia papillosa* (Linnaeus, 1767) | Sanremo | | | |
| Actinopterygii | *Anthias anthias* (Linnaeus, 1758) | Sanremo | | | LC |
| | *Callanthias ruber* (Rafinesque, 1810) | Sanremo | | | LC |
| | *Conger conger* (Linnaeus, 1758) | Sanremo | | | LC |
| | *Coris julis* (Linnaeus, 1758) | Sanremo | | | LC |
| | *Epinephelus marginatus* (Lowe, 1834) | Bordighera | III | III | VU |
| | *Labrus mixtus* Linnaeus, 1758 | Sanremo | | | LC |
| | *Mullus barbatus barbatus* Linnaeus, 1758 | Sanremo | | | LC |
| | *Muraena helena* Linnaeus, 1758 | Sanremo | | | LC |
| | *Phycis phycis* (Linnaeus, 1766) | Sanremo | | | LC |
| | *Polyprion americanus* (Bloch & Schneider, 1801) | Sanremo | | | NT |
| | *Scorpaena notata* Rafinesque, 1810 | Sanremo | | | LC |
| | *Scorpaena scrofa* Linnaeus, 1758 | Bordighera, Sanremo | | | LC |
| | *Serranus cabrilla* (Linnaeus, 1758) | Sanremo | | | LC |
| | *Spicara* sp. | Sanremo | | | LC |
| | *Trachurus* sp. | Sanremo | | | LC |

[1] Taxa found as epibiont on *E. verrucosa*.

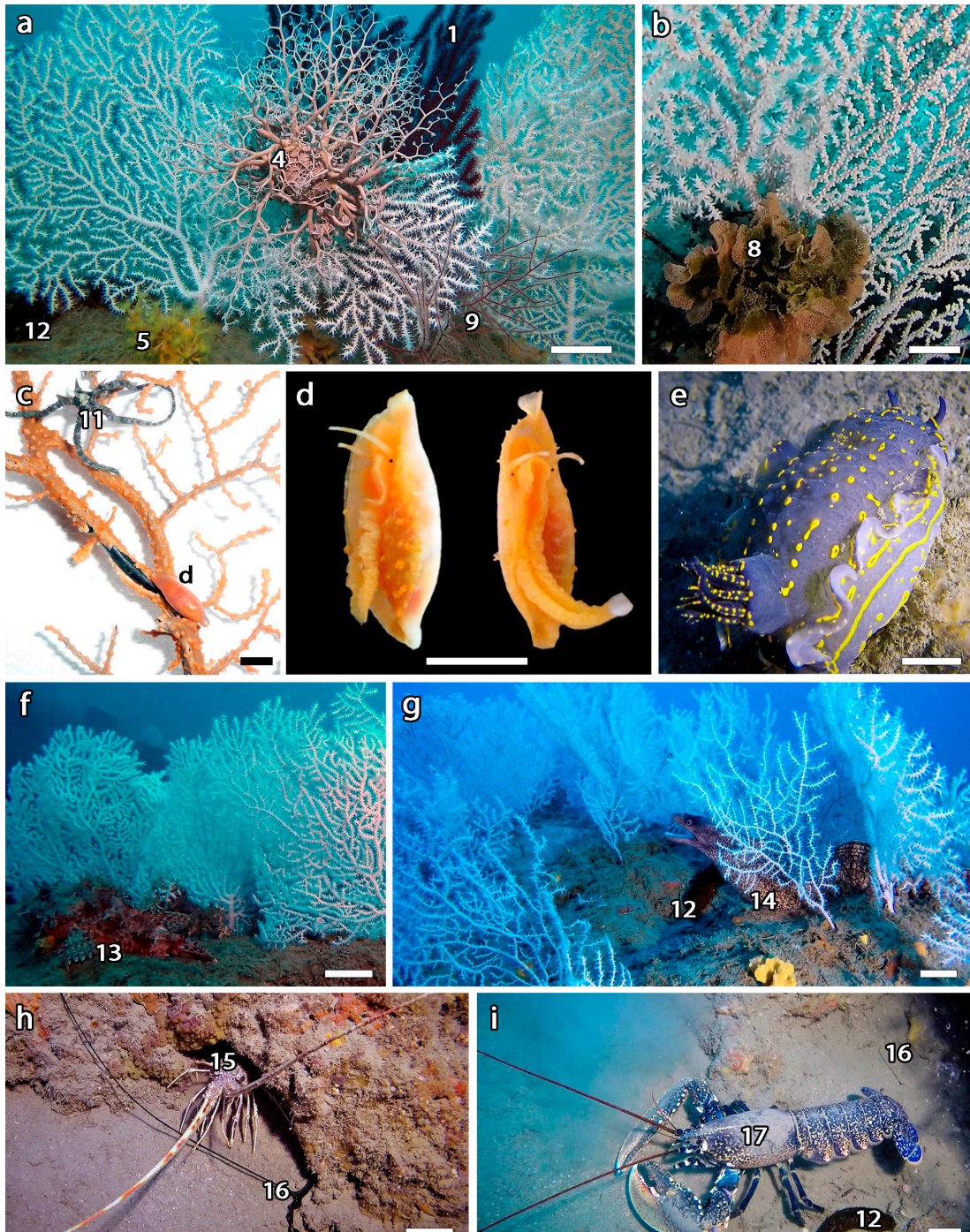

**Figure 6.** Examples of the animal taxa associated with *Eunicella verrucosa*. (**a**) Colonies of *E. verrucosa* with *Paramuricea clavata* (1) in the background, *Leptogorgia sarmentosa* (9), *Astrospartus mediterraneus* (4), *Axinella* sp. with *Parazoanthus axinellae* (5), and *Holothuria* sp. (12); (**b**) colonies of the bryozoan *Reteporella* cf. *grimaldii* (8) as epibiont on *E. verrucosa*; (**c**) sample of *E. verrucosa* with the ophiuroid *Ophiacantha setosa* (11) and the gastropod *Simnia spelta*; (**d**) lateral (left) and ventral (right) view of *S. spelta*; (**e**) the nudibranch *Felimare picta* crawling on the hard bottom below the *E. verrucosa* forest; (**f**) the red scorpionfish *Scorpaena scrofa* (13); (**g**) the Mediterranean moray eel *Muraena helena* (14); (**h**) the spiny lobster *Palinurus elephas* (15) and the echiurid *Bonnellia viridis* (16); (**i**) a large specimen of the European lobster *Homarus gammarus* (17) with *Holothuria* sp. (12) and *B. viridis* (16). Scale bars: a–b,f–g, 5 cm; c–e, 1 cm; h–I, 10 cm.

## 4. Discussion

Coral habitats play a fundamental structural role in the Mediterranean Sea, creating some of the most rich and diverse habitats known from shallow to deep bathymetries [3,4,6,8,9,16,26,27,35,48–51]. These habitats are of primary importance for marine conservation [1] but often proper protection initiatives cannot be applied due to the lack of knowledge about the presence and the distribution of the key structural species [52]. Citizen science projects based on effective and shared protocols can represent a promising source of additional information about the distribution of conspicuous species [23,46], particularly in the case of iconic taxa such as alcyonaceans. This is also the case of the pink sea fan, *E. verrucosa*, that has been considered a rare or uncommon gorgonian in the Mediterranean Sea [22,23].

### 4.1. Morphometry and Age Inference

Non-invasive methods to assess population structure on octocorals are effective and needed ways to study these peculiar communities without affecting them [7,12,22,39,53–55]. Morphometric analysis on *E. verrucosa* off Sanremo seemed to indicate a healthy and balanced population, with the symmetric distribution of both large, old colonies and juvenile, new recruits (Figure 2), also supported by the fact that signs of stress or diseases were not observed. Considering that the correlation between size and age is not linear, but follows a von Bertalanffy growth function where growth rate decreases with colonies' height [22], the inferred age-frequency distribution was in accordance with such observation, highlighting the dominant presence of colonies from 1 to 20 years old with a few colonies older than 35 years (right tail) (Figure 3). The function used to infer the age was developed from colonies in Riou Archipelago (France), close enough to Sanremo to be considered comparable. However, it was based on 15 colonies of sizes between 5 and 40 cm [22], so it should be tested and probably improved in the future, particularly with a larger number of colonies and considering also those higher than 40 cm, for which the reliability is not yet known. Whatever the maximum age of the colonies was, the presence of large and old ones indicated a certain stability of the environment and the longevity of the species. In fact, the growth rate of *E. verrucosa* in the Mediterranean Sea is $3.33 \pm 0.61$ cm year$^{-1}$ for colonies smaller than 15 cm, decreasing gradually up to $0.62 \pm 0.22$ cm year$^{-1}$ around 40 cm in height [22], and it is expected to be even lower in higher colonies. Such decrease in growth rate according to age is common in gorgonians (e.g., [56–58]). Moreover, Mediterranean oligotrophic waters can lead to an even slower growth rate of *E. verrucosa* compared to Atlantic areas [22,53], where the species is also shallower.

The population of *E. verrucosa* reported in literature at Riou Archipelago (20–40 m depth) was dominated by colonies of 10–30 cm in height, with the maximum size of 40 cm [22]. On the contrary, the more frequent size classes off Sanremo (65–70 m depth) were between 21 and 35 cm, with the highest colony of 58 cm, supporting the hypothesis that deeper populations could be larger in size (and older) than shallow ones.

### 4.2. Eunicella verrucosa in the Mediterranean Sea

On a local scale, the distribution of the colonies is usually patchy and shoals very close to each other can show different *facies*, as with the two rocky areas off Sanremo, with the two separated monospecific forests of *P. clavata* and *E. verrucosa*. On a larger scale, the data analyzed in this study showed how *E. verrucosa* is particularly widespread along the northwestern Mediterranean Sea (Figure 5a,b). However, its remarkable presence in this area of the basin could be biased by the fact that citizen science projects are more common along the coasts of Spain, France, and Italy than in the eastern basin and the coast of North Africa. Moreover, these projects are limited to scuba diving depths (up to 40–50 m), while a large part of the bathymetric range of this species falls in the lower mesophotic zone (up to 200 m), a zone still scantly explored. For this reason, citizen science campaigns in underrepresented areas and mesophotic explorations using indirect methods are needed to fill the gap of knowledge

about the geographic and bathymetric distribution of *E. verrucosa* and its forests. This approach would help to set priorities for management and conservation purposes, as already done throughout the southwest UK coastal waters where *E. verrucosa* has been used as indicator of the spatial efficacy of Marine Protected Areas [20]. Although observations of isolated colonies and occasional forests of *E. verrucosa* occurred along the Liguro-Provencal coast, technical diving explorations below the depth of classic scuba diving allowed identifying two new forests of *E. verrucosa* in the area (Figure 5b, Table 1), confirming the importance of such areas for the presence of mesophotic coral forests. Despite the need for further validation through visual surveys, those off and southeast of Kotor Bay (Figure 5a) are the only records of *E. verrucosa* forests along the coasts of Montenegro and in the whole Adriatic Sea thus far. Similarly, the occurrence of few colonies off the Merlera Island is the first record for the Ionian coasts of Greece and one of the few in the Ionian Sea (Figure 5a).

The overall biodiversity observed in association with the forests of *E. verrucosa* highlighted the importance of these habitats for both conservation and fishery management. In particular, 31 of the taxa found are listed in at least one of the main legal instruments for species conservation and management ongoing in the Mediterranean Sea, such as the Bern Convention (Convention on the Conservation of European Wildlife and Natural Habitats: Appendix II: strictly protected fauna species, Appendix III: protected fauna species), the Barcelona Convention (Protocol for Specially Protected Areas and Biological Diversity in the Mediterranean, SPA/BD: Annex II: list of endangered or threatened species, Annex III: list of species whose exploitation is regulated) and the Red List of the International Union for Conservation of Nature (IUCN) (Table 2).

Part of the megafauna associated with *E. verrucosa* forests is of high commercial value, such as the spiny lobster *Palinurus elephas* (Figure 6h), the European lobster *Homarus gammarus* (Figure 6i), and the spinous spider crab *Maja squinado*, all observed off Sanremo. Due to the relatively common presence of these crustaceans, this shoal is known as "Scoglio dell'Astice", that literally means "lobster shoal" with particular reference to *H. gammarus* whose big specimens could be present (Figure 6i). Economically important fish included *S. scrofa, M. barbatus, P. phycis, C. conger, M. helena, E. marginatus,* and *P. americanus*. This latter record is remarkable also due to the relatively rare records of this deep species at mesophotic depths in the Mediterranean Sea.

The observation of pelagic species highlighted the pivotal role of coral forests, such as *E. verrucosa* ones, as feeding ground and shelter also for species that are not strictly benthonic. The importance role of *E. verrucosa* as habitat former was also enhanced by the numerous organisms that live and feed directly on the coral colony. Some of these epibionts, such as *B. gasti, T. nilsodhneri,* and *S. spelta*, are also adapted to camouflage with *E. verrucosa* colonies.

## 4.3. Impacts and Threats

The slow growth rate of *E. verrucosa* enhances its vulnerability to anthropic pressures, particularly fishery, pollution, and climate changes. Although new to science, the *E. verrucosa* forest off Sanremo is already affected by fishing practices using bottom longlines targeting large fish, and trammel nets for lobsters. Despite being considered as passive and selectively geared with a low impact, longlines and trammel nets can sweep the seabed during retrieval operations and during bad-weather conditions. During this process they might injure or catch the corals as well as, in some cases, remain entangled on the seabed [4,34,48,59–64]. For this reason, the use of bottom-contact fishing gear should be banned in those areas characterized by vulnerable marine ecosystems as coral forests [1,65].

Corals are more susceptible to diseases when stressed. *Eunicella* species have been affected by mass mortality events linked to positive thermal anomalies [66], and evidences of a disease affecting *E. verrucosa* have been correlated to high concentrations of *Vibrio* bacteria most likely due to the elevated seawater temperature [67]. Similarly to other temperate and cold-water corals, this species could be severely affected by the ongoing climate change, with drastic habitat loss that can lead to local extinctions with limited *refugia* and negative effects on the associated community [68]. Besides the several species of commercial importance associated with the forests of *E. verrucosa*, the quantity of taxa included in

conservation frameworks is remarkable and provides an insight about the importance of this habitat for biodiversity conservation. Such diversity is due to the presence of hard-bottom and coralligenous communities [13,24–26] and a noticeable epibiotic fauna, as well as soft-bottom species that can have access to the understory of the coral forest when the hard bottom is not particularly structured.

## 5. Conclusions

The general picture about the distribution of *E. verrucosa* in the Mediterranean Sea revealed many new occurrences, although large populations constituting coral forests are still scantly known. This study represents the first quantitative study on a rare, monospecific forest of *E. verrucosa* and its remarkable associated community. Considering its accessibility and the presence of species of high commercial value, this unique and vulnerable habitat should be included in conservation strategies aiming to preserve its integrity and its biodiversity from current and potential anthropogenic threats. Examples of effective initiatives to preserve *E. verrucosa* forests and other vulnerable marine ecosystems can be the institution of Marine Protected Areas, Fishery Restricted Areas, and, in European waters, the individuation of Sites of Community Importance.

Albeit *E. verrucosa* is a shallow-water species in the Atlantic Ocean, its mesophotic occurrence in the Mediterranean Sea implies that more findings could occur in the future below 50 m of depth. The exploration of the mesophotic zone is fundamental to identify peculiar habitats such as coral forests, in order to design proper protection strategies on a basin-scale, aiming to reach a desirable network of protected areas.

**Funding:** This research was funded by the Italian Ministry of Education, University and Research (Ministero dell'Istruzione, dell'Università e della Ricerca; Programma Operativo Nazionale - PON 2014-2020), grant AIM 1807508-1, Linea 1, by the Italian Ministry for Environment, Land and Sea Protection (Ministero dell'Ambiente e della Tutela del Territorio e del Mare), as part of the Italian monitoring program for the implementation of the Marine Strategy Framework Directive (European Union, 2008/56/EC), and by the European Union through the FP7 CoCoNet project (*Towards Coast to Coast Networks of marine protected areas (from the shore to the high and deep sea), coupled with sea-based wind energy potential)*), grant number 287844.

**Acknowledgments:** I am grateful to Maurizio Delfini for underwater videos and photos, as well as to all the contributors in the citizen science platforms and projects. Special thanks to Francesco Mastrototaro for the critical suggestions to improve the clarity of the text. Chief scientists Marco Taviani, Lorenzo Angeletti, and Federica Foglini, as well as the captain, crew, and colleagues aboard R/V *Urania* during cruises CROMA and COCOMAP14 are warmly acknowledged.

**Conflicts of Interest:** The author declare no conflict of interest. The funders had no role in the design of the study; in the collection, analyses, or interpretation of data; in the writing of the manuscript, or in the decision to publish the results.

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
