# Peer review of "Vulnerable Forests of the Pink Sea Fan Eunicella verrucosa in the Mediterranean Sea"

_diversity, doi:10.3390/d12050176_

Round 1
Reviewer 1 Report
Abstract
It is well written and summary the content of the manuscript.
Introduction
Line 38 Please check this sentence “ It dwells in the mesophotic zone”
Line 52 please add “,” after Croatia.
Line 56-58 I suggest to make this sentence more shorten “
Materials and Methods
Line 99 I suggest to the author to eliminate this sentence “i.e. a social network of nature enthusiasts that share and cross-validate photographic observations”
Line 100 I suggest to the author to eliminate this sentence “based on an international network to share open access data about life”
Line 103 I suggest to the author to eliminate this sentence “that groups underwater observations carried out by trained volunteers 103 based on a standardized protocol”
Line 113 I suggest to add also the name “Scoglio dell’Astice”
Line 118 please change “present” with “observed”
Figure 1 please change “Benthic community off Sanremo’ with “Benthic communities off Sanremo”
3.2. The coral forest off Bordighera I suggest to the author to merge this paragraph with 3.1 and rename the paragraph title.
Have the author density values of the anthropogenic impacts detected in the two sites (apparently missing for Bordighera site)? Is it possible to define which it is highly impacted?
I also suggest to take separate the images showing author propriety data from those related to additional data. Maybe the most easy choice is to insert the images related to the Bordighera site in the Figure 3.
Line 160 I suggest to change “Albeit it” with “Even if”
3.5. Biodiversity associated with Eunicella verrucosa forests
Line 230-244 In my opinion this last part might be moved in discussion section
In particular, the author could consider to insert this part in the second suggest paragraph (i.e. “threats associated with E. verrucosa” see the suggestion to the Discussion Section).
Discussion
I suggest to the author to divide the Discussion section in two paragraphs one focused on the distribution and comparison of the E. verrucosa with the additional sites, and another paragraph focused on the threats associated with this coral.
Regarding the first suggested paragraph, Can the author propose some ecological hypothesis about the almost lack of this species in the eastern Mediterranean basin?
Regarding the second suggested paragraph I suggest to the author to include some considerations regarding the importance of the establishment of a Marine Protected Area.
Finally, the author need to add reference letter (for example Fig 2a, or b,c,d,….) to the cited figure in the text.
Author Response
I thank the Reviewer for the interesting comments and suggestions. I re-organized the text accordingly. Please find here a point-by-point response.
Introduction
Line 38 Please check this sentence “ It dwells in the mesophotic zone”
Response: Sentence corrected as follow: “This species lives in the mesophotic zone”
Line 52 please add “,” after Croatia.
Response: Done.
Line 56-58 I suggest to make this sentence more shorten “
Response: Done.
Materials and Methods
Line 99 I suggest to the author to eliminate this sentence “i.e. a social network of nature enthusiasts that share and cross-validate photographic observations”
Response: I prefer to keep the sentence because many readers can be unaware about what iNaturalist is and why I chose this platform. The purpose is to highlight that the data used are reliable and can be verified. I put the information in brackets to increase the readability of the sentence.
Line 100 I suggest to the author to eliminate this sentence “based on an international network to share open access data about life”
Response: I prefer to keep the sentence for the same reason explained for iNaturalist.
Line 103 I suggest to the author to eliminate this sentence “that groups underwater observations carried out by trained volunteers 103 based on a standardized protocol”
Response: I prefer to keep the sentence for the same reason explained for iNaturalist.
Line 113 I suggest to add also the name “Scoglio dell’Astice”
Response: Done.
Line 118 please change “present” with “observed”
Response: Done.
Figure 1 please change “Benthic community off Sanremo’ with “Benthic communities off Sanremo”
Response: Done.
3.2. The coral forest off Bordighera
I suggest to the author to merge this paragraph with 3.1 and rename the paragraph title.
Response: unfortunately the forest off Bodighera was not studied with the same detail of the one off Sanremo, due to logistic problems. For this reason I prefer to keep the paragrahs separated to avoid confusion. In fact, the focus of the study is off Sanremo, because it is the only population that was quantified in terms of density and size-frequency distribution, as explained in the methods. Moreover, the two sites are slightly different, because off Sanremo the coral forest is monospecific, while off Bordighera it is mixed: one more reason to keep them separated and avoid confusion.
Have the author density values of the anthropogenic impacts detected in the two sites (apparently missing for Bordighera site)? Is it possible to define which it is highly impacted?
Response: Unfortunately I have not enough data to quantify the impacts in the site off Bordighera. The information currently available is not strong enough to support a proper comparison between the two sites or with other sites.
I also suggest to take separate the images showing author propriety data from those related to additional data. Maybe the most easy choice is to insert the images related to the Bordighera site in the Figure 3.
Response: I want to clarify that I have the property of all the images. Underwater ones have been done by a collaborator of mine, others by myself. The logic behind the figures is that figure 3 is focused on the site off Sanremo (Scoglio dell’Astice) that has been surveyed with a great detail. Figure 4 concerns other new occurrences of E. verrucosa here presented but not yet studied in detail. For this reason I prefer to keep the figures as they are, to avoid misunderstandings.
Line 160 I suggest to change “Albeit it” with “Even if”
Response: Done.
3.5. Biodiversity associated with Eunicella verrucosa forests
Line 230-244 In my opinion this last part might be moved in discussion section
In particular, the author could consider to insert this part in the second suggest paragraph (i.e. “threats associated with E. verrucosa” see the suggestion to the Discussion Section).
Response: Done.
Discussion
I suggest to the author to divide the Discussion section in two paragraphs one focused on the distribution and comparison of the E. verrucosa with the additional sites, and another paragraph focused on the threats associated with this coral.
Response: I agree. Based on the new re-organization of the text, I identified three paragraphs, to discuss in order the results:
4.1. Morphometric and age inference
4.2. Eunicella verrucosa in the Mediterranean Sea
4.3. Impacts and threats
Regarding the first suggested paragraph, Can the author propose some ecological hypothesis about the almost lack of this species in the eastern Mediterranean basin?
Response: I prefer not to add further hypothesis in the discussions. To be honest, I am not even sure that the species is lacking in the eastern basin, but probably studies and surveys have been less in the eastern Mediterranean compared to the Western. For this reason in the text I say “However, its remarkable presence in this area of the basin could be biased by the fact that citizen science projects are more common along the coasts of Spain, France and Italy than in the eastern basin and the coast of North Africa. Moreover, these projects are limited to scuba diving depths (up to 40–50 m), while a large part of the bathymetric range of this species falls in the lower mesophotic zone (up to 200 m), a zone still scantly explored. For this reason, citizen science campaigns in underrepresented areas and mesophotic explorations using indirect methods are needed to fill the gap of knowledge about the geographic and bathymetric distribution of E. verrucosa and its forests”.
Regarding the second suggested paragraph I suggest to the author to include some considerations regarding the importance of the establishment of a Marine Protected Area.
Response: I agree, but I think that it fits more in the conclusion part. I added in the conclusions the following sentences “Considering its accessibility and the presence of species of high commercial value, this unique and vulnerable habitat should be included in conservation strategies aiming to preserve its integrity and its biodiversity from current and potential anthropogenic threats. Examples of effective initiatives to preserve E. verrucosa forests and other vulnerable marine ecosystems can be the institution of Marine Protected Areas, Fishery Restricted Areas and, in European waters, the individuation of Sites of Community Importance”.
Finally, the author need to add reference letter (for example Fig 2a, or b,c,d,….) to the cited figure in the text.
Response: Done.
Reviewer 2 Report
This study presents a contribution to the current information available about the distribution and associated fauna of Eunicella verrucosa in the Mediterranean Sea. The here reported new records of this species in the Ligurian and Adriatic Sea were added to the currently available information about the distribution of this species through Citizen Science and reports in scientific literature, resulting in the most recent update of its distribution in the Mediterranean. Furthermore, the identification of the associated fauna showed that E. verrucosa forests host a series of species of conservation and commercial interest, thus highlighting the importance of protecting these habitats through proper conservation management.
My comments and suggestions are mostly related to text editing. Mainly, I would suggest combining the RESULTS and DISUCSSION sections into one, especially as the RESULTS subsection “3.5 Biodiversity associated with Eunicella verrucosa forests” already reads like both sections have been combined. In case the author decides against it, then either the RESULTS or the DISCUSSION section should be reorganized to make them consistent regarding the order the results are presented and discussed (e.g., the DISCUSSION starts with the results on the combined records for the Mediterranean, followed by discussing the here found new records, while the inverse order is shown in the RESULTS). Also, I recommend a thorough revision of the English. In general terms it is okay, but there are many minor errors throughout the text.
Specific Comments:
- The Figures cited in the text should be specified, e.g. Fig. 1a or Fig. 1b, etc…
- Species names in Legend of Figure 6 should be in cursive.
- Line 279-282: This sentence should be in the Conclusion. Also, to support his suggestion, the author could cite here the work of Pikesley et al. (2016), “e.g., such as already being applied for E. verrucosa surveys by the "Seasearch" programme throughout southwest UK coastal waters.”
Pikesley, S. K., Godley, B. J., Latham, H., Richardson, P. B., Robson, L. M., Solandt, J. L., ... & Witt, M. J. (2016). Pink sea fans (Eunicella verrucosa) as indicators of the spatial efficacy of Marine Protected Areas in southwest UK coastal waters. Marine Policy, 64, 38-45.
- Line 327-332: This paragraph seems somehow unrelated to and out of place in the discussion. I suggest removing it.
- Line 414: Correct the first author’s name to “Sartoretto”
Author Response
I thank the Reviewer for the interesting comments and suggestions. I accepted all of them, and rearranged the text accordingly. Please find here a point-by-point response.
My comments and suggestions are mostly related to text editing. Mainly, I would suggest combining the RESULTS and DISUCSSION sections into one, especially as the RESULTS subsection “3.5 Biodiversity associated with Eunicella verrucosa forests” already reads like both sections have been combined. In case the author decides against it, then either the RESULTS or the DISCUSSION section should be reorganized to make them consistent regarding the order the results are presented and discussed (e.g., the DISCUSSION starts with the results on the combined records for the Mediterranean, followed by discussing the here found new records, while the inverse order is shown in the RESULTS). Also, I recommend a thorough revision of the English. In general terms it is okay, but there are many minor errors throughout the text.
Response: I prefer to keep separated Results and Discussions, but I agree in the need for a more clear setting of the discussions that follows the results. Based on the new re-organization of the text, I identified three paragraphs:
4.1. Morphometric and age inference
4.2. Eunicella verrucosa in the Mediterranean Sea
4.3. Impacts and threats
English has been revised by a native-English lecturer.
Specific Comments:
- The Figures cited in the text should be specified, e.g. Fig. 1a or Fig. 1b, etc…
Response: Done.
- Species names in Legend of Figure 6 should be in cursive.
Response: Done.
- Line 279-282: This sentence should be in the Conclusion. Also, to support his suggestion, the author could cite here the work of Pikesley et al. (2016), “e.g., such as already being applied for E. verrucosa surveys by the "Seasearch" programme throughout southwest UK coastal waters.”
Pikesley, S. K., Godley, B. J., Latham, H., Richardson, P. B., Robson, L. M., Solandt, J. L., ... & Witt, M. J. (2016). Pink sea fans (Eunicella verrucosa) as indicators of the spatial efficacy of Marine Protected Areas in southwest UK coastal waters. Marine Policy, 64, 38-45.
Response: I agree. Modified as follow: “...citizen science campaigns in underrepresented areas and mesophotic explorations using indirect methods are needed to fill the gap of knowledge about the geographic and bathymetric distribution of E. verrucosa and its forests. This approach would help to set priorities for management and conservation purposes, as already done throughout the southwest UK coastal waters where E. verrucosa has been used as indicator of the spatial efficacy of Marine Protected Areas [20]”
- Line 327-332: This paragraph seems somehow unrelated to and out of place in the discussion. I suggest removing it.
Response: I prefer to keep it, because otherwise it seems that fishing impacts are the only threats to coral conservation. On the contrary, climate changes needs to be mentioned among the threats and human impacts since comprehensive management actions should not ignore the importance of initiatives to mitigate climate changes.
- Line 414: Correct the first author’s name to “Sartoretto”
Response: Done.
Round 2
Reviewer 1 Report
Dear author the manuscript has been improved. In my opinion it can be accepted.